# Impact of Smoking on Neutrophil Enzyme Levels in Gingivitis: A Case-Control Study

**DOI:** 10.3390/ijerph18158075

**Published:** 2021-07-30

**Authors:** Rumeysa Omer-Cihangir, Ulku Baser, Canan Kucukgergin, Gokce Aykol-Sahin, Olivier Huck, Funda Yalcin

**Affiliations:** 1Periodontology Department, Faculty of Dentistry, University of Istanbul, Istanbul 34093, Turkey; fyalcin@istanbul.edu.tr; 2Department of Medical Biochemistry, Istanbul Faculty of Medicine, University of Istanbul, Istanbul 34093, Turkey; ckgergin@istanbul.edu.tr; 3Periodontology Department, Faculty of Dentistry, Istanbul Okan University, Istanbul 34959, Turkey; gokce.aykol@okan.edu.tr; 4Department of Periodontology, Dental Faculty, University of Strasbourg, 67081 Strasbourg, France; huck.olivier@gmail.com; 5INSERM, UMR 1260 Osteoarticular and Dental Regenerative Nanomedicine, 67000 Strasbourg, France; 6Pôle de Médecine et de Chirurgie Bucco-Dentaires, Hôpitaux Universitaires de Strasbourg, 67091 Strasbourg, France

**Keywords:** gingival crevicular fluid, dental plaque, biofilm, gingivitis, myeloperoxidase, glucuronidase, neutrophil elastase, neutrophil enzymes, smoking

## Abstract

Background: The determination of the impact of risk factors such as smoking in periodontal disease development is of importance to better characterize the disease. However, its impact on host response remains unclear. This study aimed to evaluate the effects of tobacco smoking on GCF levels of neutrophil enzymes (myeloperoxidase (MPO), beta-glucuronidase (BGD), neutrophil elastase (NE) and periodontal parameters in healthy young adults with dental plaque biofilm-induced gingivitis. Methods: The study population consisted of 60 systemically healthy young adults (39 smokers (Sm) and 21 non-smokers (n-Sm)) diagnosed with plaque-induced gingivitis. The periodontal examination consisted of a plaque index (PI); gingival index (GI); probing depth (PD); bleeding on probing (BoP), and clinical attachment level (CAL). GCF MPO, BGD, and NE levels were determined by means of an enzyme-linked immunosorbent assay (ELISA). Results: PI, GI, and BoP were significantly increased in the Sm group (*p* < 0.05). PD and CAL showed no significant difference between Sm and *n*-Sm groups (*p* > 0.05). In GCF, MPO, BGD, and NE levels were significantly increased in Sm group (*p* < 0.05). NE levels showed a significant correlation with GI and BoP (*p* < 0.05 for both). Moreover, a positive correlation between BGD and NE levels (*p* < 0.05) was measured. Conclusions: It may be concluded that, even in young patients, tobacco consumption affects the host’s immune response related to gingival inflammation. It is, therefore, mandatory to inform young patients about the risk related to tobacco consumption for their gingival health.

## 1. Introduction

Gingivitis is an inflammatory disease that is triggered by the accumulation of oral biofilms [1] and considered to be the precursor stage of periodontitis. The management of gingivitis is, therefore, instrumental in the prevention of periodontitis [2]. In gingivitis, both bacterial insult and host-response are keys in the initiation and development of the inflammatory process. At each step of gingivitis establishment, the recruitment and activation of neutrophils has been observed [3]. Indeed, neutrophils are key in periodontal health but also in the initiation phase of the lesion and in the progression of the disease, both being associated with exacerbated neutrophil activation [4,5]. Such sustained neutrophils activity promotes the accumulation of helper T cells, B lymphocytes, and plasma cells in the established and advanced lesion [6].

Neutrophils have several defense mechanisms. Intracellular defense is associated with phagocytosis and extracellular defense includes degranulation and neutrophil extracellular traps. Degranulation is the process followed by the release of antimicrobial cytotoxic molecules in granules and secretory vesicles as a result of hypoxia following antigenic stimulation [7]. Myeloperoxidase (MPO), beta-glucuronidase (BGD), and neutrophil elastase (NE) are proteolytic enzymes released from azurophilic granules of neutrophils into the extracellular area following neutrophil stimulation [8]. MPO is a hemeprotein that constitutes at least 5% of the dry weight of leucocytes. This enzyme is responsible for the generation of reactive oxygen species in the peroxidation and halogenation cycles, and with the oxidative potential of its co-substrate, hydrogen peroxide, it exhibits bactericidal activity [9]. BGD is a lysosomal acid hydrolase that plays an active role in the degradation of proteoglycans in connective tissue [10]. NE is a serine protease that is released from azurophilic granules during phagocytosis and cell lysis. It acts against elastin, collagen, fibronectin, hemoglobin, laminin 1, proteoglycans, and thrombospondin [11]. NE also allows neutrophil transmigration. Excessive secretion of NE has the potential to lead to extracellular matrix degradation [12]. Therefore, enzymatic activity takes an important place in neutrophil defense and through all process of the periodontal disease including initiation and progression from gingivitis to advanced stages of periodontitis. Gingival crevicular fluid (GCF) levels of these enzymes were reported to be increased in periodontitis suggesting that MPO, BGD, and NE may be potential markers of periodontal inflammation [13].

Previous studies showed that the immune-inflammatory response may be affected by tobacco smoking, an important risk factor of periodontal disease, causing impaired neutrophil functions, elevated production of inflammatory cytokines, delayed revascularization, decreased production of collagen, and increased production of collagenase [5,6,14]. Tobacco smoking is considered one of the major periodontal risk factors among others such as metabolic factors, pharmacological agents, hematological conditions, nutritional factors, elevations in sex hormones, dental plaque biofilm retention factors, and oral dryness [2], since it was found to be the major environmental risk factor for periodontal disease [15,16]. It has been reported that oral and peripheral neutrophils, which are exposed to nicotine, display impaired neutrophilic activities including impaired phagocytosis, suppressed production of protease inhibitors, and deteriorated enzymatic activity, suggesting that smoking may induce the conversion of periodontal homeostasis into a progressive disease [17,18]. GCF levels of MPO and NE were reported to be increased in smokers (Sm) with periodontitis [19,20,21]. Although it is known that smoking has effects on neutrophils enzymatic activity, the effect of smoking on GCF levels of MPO, BGD, and NE in patients with gingivitis was not thoroughly investigated.

Several clinical studies indicate that gingival inflammation and bleeding on probing (BoP) are less pronounced in smokers [22,23]. Contrariwise, several studies have reported that smokers exhibit more prevalent BoP and plaque as compared to non-smokers [24,25]. This may be due to enhanced blood pressure caused by smoking and defused vasoconstrictive effects of nicotine [26,27]. Moreover, periodontal status was proven to be influenced by the number of cigarettes smoked and smoking duration [28].

Currently, divergent results regarding the correlation between clinical parameters and GCF levels of MPO, BGD, and NE in periodontal disease are available. Some studies reported limited correlations between these variables [19,29,30], whereas some others [31,32] markedly correlated GCF enzyme levels to clinical parameters.

In this study, we hypothesized that smoking may affect neutrophil enzyme levels in GCF leading to more severe clinical periodontal inflammation. Therefore, the present study was designed with the following aims: (1) to evaluate the effects of tobacco consumption on the degree of gingival inflammation in a sample of young healthy adults; and (2) to compare MPO, BGD, and NE levels in GCF among Sm and *n*-Sm.

## 2. Materials and Methods

### 2.1. Study Population

Patients attending the Department of Periodontology, Faculty of Dentistry, Istanbul University, for periodontal consultation were considered to be included in this study. A total of 60 healthy subjects (38 men and 22 women; mean age 21.46 ± 2.39 yo, range 18 to 25 years) diagnosed with dental plaque biofilm-induced gingivitis were included in the study. Patients who reported smoking more than 5 cigarettes a day for at least 12 months [33] were allocated to Sm group. Ethical approval for the study was obtained from the Istanbul University, Faculty of Dentistry Ethics Committee (File Number: 2019/43). Signed written informed consents were received from every subject. Detailed dental and medical anamnesis were recorded for every subject.

### 2.2. Including and Excluding Criteria

Inclusion criteria for all subjects were as follow: 18–25 yo, diagnosis of dental plaque biofilm-induced gingivitis defined as ≥10% of sites with BoP and mean probing depth (PD) ≤3 mm [2]. Exclusion criteria for all subjects were as follow: systemic diseases, diagnosis of periodontitis or having been treated for periodontal disease during the last 6 months, pregnancy, use of antibiotics, steroidal, or non-steroidal anti-inflammatory drugs in the 6 months prior to the study.

### 2.3. Determination of Clinical Periodontal Status

To determine the clinical periodontal status, gingival Index (GI), plaque Index (PI), probing depth (PD), bleeding on probing (BoP), and clinical attachment level (CAL) were recorded at mesiobuccal, midbuccal, distobuccal, mesiolingual, midlingual, and distolingual sites of each tooth using a PCPUNC15 periodontal probe (Hu Friedy, Chicago, IL, USA).

### 2.4. Sampling of Gingival Crevicular Fluid

Prior to each periodontal examination, GCF samples were collected from distal sites of maxillary right canines and mandibular left canines with Periopaper^®^ strips (Ora Flow, Plainview, NY, USA) according to the method described by Rüdin et al. [34]. Before collecting GCF, the sampling sites were gently air-dried and kept dry with cotton wool rolls. The tip of each strip was placed, avoiding any blood contamination, into the gingival sulcus for 30 s and placed into a 1.5 mL plastic Eppendorf^®^ tube (Hamburg, Germany) covered with Parafilm™ stretches (Sigma-Aldrich Chemble GMBH, Delsenhofen, Germany) and kept at −80 °C until assayed. Strips and tubes were weighed together before and after collecting GCF using Sartorius^®^ A200S high-precision scale (Göttingen, Germany). The weight of the fluid of each strip, expressed in μg, was converted to volume in μL by assuming that the density of GCF was 1.0 mg/mL [35].

### 2.5. Determination of Enzyme Levels

First, GCF was eluted from the paper strips by adding 300 μL Bovine Serum Albumin-Phosphate-Buffered Saline (BSA-PBS) to each tube. The tubes were left on shaking platform for 20 h at 4 °C. Then, GCF levels of MPO, BGD, and NE were measured using commercially available ELISA kits (Thermo Fisher, Vienna, Austria kits for MPO and NE, MyBioSource, Shanghai, CHINA kit for BGD) according to the manufacturer’s instructions. The absorbance was measured at 450 nm using a spectrophotometer plate reader (Eon-Biotek, Winooski, VT, USA). The detection sensitivity for MPO was 0.026 ng/mL, for BGD was 1.0 U/mL and for NE was 1.98 ng/mL. Results were expressed as μg/mL for MPO and NE and U/μL for BGD.

### 2.6. Statistical Analysis

Data are expressed as means, medians, and standard deviations (SD). The findings of previous studies where [19] NE were measured and where [36] the smokers: Non-smokers ratio in gingivitis patients was found to be 2:1 were utilized for statistical power analysis. With a power of 80% and a confidence interval of 95%, minimal sample size required for the comparison was estimated to be of 20 patients with a 13:7 ratio and tripled to anticipate potential dropouts. The primary outcome was the GCF level of NE and secondary outcomes were GCF levels of MPO, BGD, and clinical parameters, which were PI, GI, BoP, PD, and CAL. Distribution normality was analyzed by Kolmogorov–Smirnov test. GI, BoP, and GCF levels of NE showed normal distribution and PI, PD, CAL, and GCF levels of MPO and BGD did not show normal distribution. The statistical significance of differences between Sm and N-Sm for categorical variables was calculated according to Chi-Square test and for non-categorical variables was calculated according to a Mann–Whitney U-test and an Independent Samples *t*-test. Correlations were determined by Pearson rank correlation coefficient. Models were constructed by defining gender, smoking status, BoP, PI, PD, CAL, and GI scores as independent variables and GCF levels of BGD as dependent variable. In order to estimate the probability of independent variables in terms of enzyme levels, binary logistic regression analysis (Forward LR) was performed. The null hypothesis was rejected at *p* < 0.05.

## 3. Results

### 3.1. Demographic Characteristics

Sm group consisted of 39 subjects (29 men and 10 women; mean age 21.28 ± 2.45) and *n*-Sm group consisted of 21 subjects (12 women and 9 men; mean age 21.80 ± 2.29). Regarding demographic characteristics, in the Sm group, a majority of men were included (*p* < 0.05 vs. *n*-Sm) while no significant differences were observed between groups for age, education distribution, and frequency of tooth-brushing (*p* > 0.05) (Table 1). In Sm group, the mean duration of smoking and daily consumption of cigarettes were 4.76 ± 3.50 years and 14.74 ± 0.77 cigarettes respectively (Table 2) with an average of 3.5 pack-years of smoking.

### 3.2. Clinical Analyses

Smokers showed significantly higher PI, GI, and BoP scores (*p* > 0.05), whereas the difference between Sm and *n*-Sm for PD and CAL failed to reach significance since the patients were all diagnosed with dental plaque biofilm-induced gingivitis (*p* > 0.05) (Table 2).

### 3.3. Biochemical Analyses

In GCF, smokers showed significantly higher levels of MPO, BGD, and NE (*p* < 0.05) (Table 3). MPO levels in GCF were measured at 288.61 ± 283.69 μg/mL and 174.83 ± 137.23 μg/mL in Sm and *n*-Sm respectively (*p* < 0.001). BGD levels in GCF were measured at 26.95 ± 2.50 U/μL and at 21.78 ± 2.95 U/μL in Sm and *n*-Sm, respectively (*p* = 0.027). Regarding NE levels, 301.24 ± 193.02 μg/mL and 176.20 ± 92.64 μg/mL in Sm and in *n*-Sm (*p* = 0.007) were measured.

### 3.4. Correlations

In this sample population consisting of individuals with dental plaque biofilm-induced gingivitis, GCF levels of MPO and BGD showed no significant association with clinical parameters (*p* > 0.05), while GCF level of NE was positively correlated with GI, BoP, and CAL (*p* < 0.05) but not with PI and PD (*p* > 0.05) (Table 4). GCF levels of MPO showed no significant correlation with GCF levels of BGD and NE (*p* > 0.05), whereas GCF levels of BGD correlated positively with GCF levels of NE (*r* = 0.321, *p* = 0.006). In smokers, MPO and BGD were not associated with clinical parameters (*p* > 0.05) (Table 4), smoking duration, and cigarette consumption (*p* > 0.05) (Table 5), while GCF levels of NE correlated positively only with GI, BoP, and CAL (*p* < 0.05) (Table 4). In non-smokers, MPO and BGD showed no statistically significant association with clinical parameters (*p* > 0.05), while GCF levels of NE correlated positively only with PI (*p* < 0.05) (Table 4).

### 3.5. Logistic Regression Analysis

Smoking status explained the model. The probability of having GCF levels of BGD above 25.34 μg/mL in Sm has an odds ratio (OR) = 20.00 time (95% CI: 4.780–83.687) (Table 5).

## 4. Discussion

The present study aimed to evaluate the effects of tobacco smoking on clinical and biochemical parameters in healthy young adults exhibiting dental plaque biofilm-induced gingivitis. Herein, the data imply that the higher levels of neutrophil enzymes in GCF in Sm seem to be associated with increased inflammation influenced by smoking.

Gingivitis is the most prevalent inflammatory disease affecting gingival tissues [37]. Although the treatment of gingivitis through oral hygiene education and supra-gingival scaling is highly effective, untreated gingivitis can progress to periodontitis [1]. Gingivitis is associated with an inflammatory reaction involving the neutrophils that are key players in the homeostasis between host and microbiota [4]. Through their strong and diverse antimicrobial mechanisms, neutrophils can cope with a wide spectrum of bacteria, fungi, and protozoa [8]. Neutrophil elastase is shown to be essential to initiate neutrophil extracellular trap formation synergized with myeloperoxidase to drive chromatin de-condensation [38].

It has been highlighted in the consensus report of the 2017 World Workshop on the Classification of Periodontal and Peri-Implant Diseases and Conditions [2] that the methods of defining dental plaque biofilm-induced gingivitis may include grading of BoP and defining percentages as mild for <10% BoP, moderate for 10–30% BoP, and severe for >30% BoP. This approach includes the extent of inflammation regardless of the degree of severity. It might help to elucidate the definition of dental plaque biofilm-induced gingivitis by including biochemical diagnosis in clinical studies. Here, we measured clinical outcomes (PI, GI, BoP, PD, and CAL) at six sites of each tooth to facilitate a better clinical interpretation and to avoid underestimation of the gingival inflammation. Our definition of gingival inflammation was based on the 2017 consensus report [2] (BoP > 10%) which includes the percentage of BoP calculated by full mouth examination. All patients were diagnosed with generalized dental plaque biofilm-induced gingivitis with a BoP ≥ 30% score. PD and CAL did not exhibit a significant difference between Sm and *n*-Sm as expected since the patients were diagnosed with dental plaque biofilm-induced gingivitis.

The descriptive data of this study demonstrated higher plaque accumulation in Sm as observed in previous studies [39,40]. It should be emphasized that smokers tend to have poor oral hygiene, as demonstrated by the significantly higher PI compared to *n*-Sm. Even though Sm have the same tooth-brushing habits as *n*-Sm, Sm present higher PI scores which could be attributed to the enhancing effect of smoking on local plaque accumulation. Machuca et al. [41] interpreted the higher PI scores in smokers, despite the same tooth-brushing habits, as a consequence of an ecological change within the biofilm. It would be proper to mention that tobacco smoking is compromising oral health-related life quality as a result of poor oral hygiene [42].

Tobacco smoking was related to the pathogenesis of periodontal disease [15,16] as well as pulmonary, cardiovascular, and gastrointestinal diseases [43]. Although limited studies are examining the relationship between smoking and periodontitis by measuring neutrophil enzyme activity [19,20,21], their role in gingivitis is described less. Here, it was observed a higher neutrophil enzymatic activity in GCF in Sm than in *n*-Sm. Production of neutrophil-derived enzymes, a crucial component of neutrophil defense, was reported to be increased in GCF in Sm [19,20,21] as observed in the present study.

The relationship between smoking and gingival inflammation is not clearly understood. Several studies have shown less pronounced gingival inflammatory symptoms, as reflected by BoP, in Sm compared to *n*-Sm [22,23]. Numerous studies have investigated the effects of nicotine on periodontal tissues and hypothesized an impact on blood flow due to its vasoconstrictive properties. However, findings indicated either no change or increased flow rate measured by laser Doppler flowmeter [26,27]. In a multicenter study, heavy smokers presented significantly lower levels of gingival inflammation, as reflected by both GI and GBI (Gingival Bleeding Index), than both light and moderate smokers [44]. Some other studies reported severe gingivitis in smokers [25,41,42,43]. Although varied studies conclude in diversified ways, they all point to significantly compromised periodontal conditions. The present study revealed that smoking has unfavorable effects on gingival inflammation. Since this study does not include a group of healthy participants, it has been only emphasized a possible association between smoking and gingivitis.

Another criterion, age, as specified in the 18–25 years range aimed to identify periodontal patients before the occurrence of irreversible periodontal destruction. Additionally, the clinical signs of inflammation are expected to be still detectable during this age range. The results showed a significant correlation between our primary outcome, GCF levels of NE, and clinical signs of gingival inflammation reflected by GI and BoP. According to the logistic regression analysis, smokers present the risk of having 2.19 times higher GCF levels of BGD. It may imply that moderate smoking (average 14 cig/day) for a comparably short duration (average 4 years) shows an association with GCF levels of neutrophil-derived proteases. Follow-up studies should be performed to confirm this observation.

## 5. Conclusions

The upregulation of GCF levels of MPO, BGD, and NE in smokers may be due to an over-activated host response. The potential neutrophil-mediated tissue injury can lead to further attachment loss and severe periodontal disease. Within the limitations of the present study, the following conclusions may be drawn; (1) elevated levels of MPO, BGD, and NE in smokers during the inflammation may contribute to compromising the periodontal support; (2) Sm tend to have poorer oral hygiene compared to *n*-Sm that negatively impact periodontal health and oral-health related quality of life, and (3) tobacco smoking should be considered a major risk factor for periodontal disease from an early age.

It is extremely important to prevent smoking at an early age and to promote better periodontal monitoring. These are considered to improve oral-health-related quality of life.

## Figures and Tables

**Table 1 ijerph-18-08075-t001:** Demographic Data for Smokers and Non-Smokers with Gingivitis.

Characteristic	Sm (*n* = 39)	*n*-Sm (*n* = 21)	Total (*n* = 60)	*p*-Value
Age (years), mean ± SD	21.28 ± 2.45	21.80 ± 2.29	21.46 ± 2.39	0.421 *
Gender, *n* (%)				0.016 ^†^
Male	29 (74.4)	9 (42.9)	38 (63.3)	
Female	10 (25.6)	12 (57.1)	22 (36.7)	
Education, *n* (%)				0.704 ^†^
<Bachelor degree	5 (12.8)	2 (9.5)	7 (11.6)	
‡ Bachelor degree	34 (87.2)	19 (90.5)	53 (88.4)	
Tooth-brushing frequency, *n* (%)				0.275 ^†^
Twice a day or more	24 (61.5)	17 (81.0)	18 (30.0)	
Once a day	14 (35.9)	4 (19.0)	41 (68.3)	
Less than once a day	1 (2.6)	0 (0.0)	1 (1.7)	
Smoking duration (year), *n* (%), mean ± SD	4.76 ± 3.50			
1–5 years, median, (IQR)	28 (71.8)			
6–10 years, median, (IQR)	8 (20.5)			
11–15 years, median, (IQR)	3 (7.7)			
Cigarettes consumption (cig/day), *n* (%), mean ± SD	14.74 ± 0.77			
5–10 cig. per day, median, (IQR)	8 (20.5)			
10–20 cig. per day, median, (IQR)	14 (35.9)			
20+ cig. per day, median (IQR)	17 (43.6)			

* Differences between groups were tested using Independent Samples *t*-test. ^†^ Association between smoking and each independent variable were analyzed using the *x*^2^ test and Tukey’s post hoc analysis. ‡ Bachelor degree or advanced.

**Table 2 ijerph-18-08075-t002:** Clinical Data for Smokers and Non-Smokers with Gingivitis.

Characteristics	Sm (*n* = 39)	*n*-Sm (*n* = 21)	Total (*n* = 60)	*p* Value
PI	1.76 ± 0.65	1.19 ± 0.5	1.56 ± 0.66	0.002 *
GI	1.73 ± 0.18	1.55 ± 0.41	1.67 ± 0.29	0.023 ^†^
BoP (%)	75.48 ± 12.12	65.33 ± 20.0	71.93 ± 15.94	0.017 ^†^
PD (mm)	1.38 ± 0.2	1.35 ± 0.17	1.37 ± 0.19	0.698 *
CAL (mm)	1.37 ± 0.64	1.34 ± 0.54	1.36 ± 0.19	0.664 *

All values presented as means ± SD. * Calculated using a Mann–Whitney U-test; *p* < 0.05. ^†^ Calculated using an Independent Samples *t*-test; *p* < 0.05. PI: Plaque Index, GI: Gingival Index, BoP: Bleeding on Probing, PD: Probing Depth, CAL: Clinical Attachment Level.

**Table 3 ijerph-18-08075-t003:** Descriptive Statistics for Biochemical Parameters of Study Participants According to Smoking.

Characteristics	Sm (*n* = 39)	*n*-Sm (*n* = 21)	Total (*n* = 60)	*p* Value
MPO (μg/mL), mean ± SD	288.61 ± 283.69	174.83 ± 137.23	248.79 ± 247.41	<0.001 *
BGD (U/μL), mean ± SD	26.95 ± 2.50	21.78 ± 2.95	25.14 ± 3.63	0.027 *
NE (μg/mL), mean ± SD	301.24 ± 193.02	176.20 ± 92.64	257.47 ± 174.71	0.007 ^†^

* Calculated using Mann Whitney U-test; *p* < 0.05. ^†^ Calculated using Independent Samples *t*-test; *p* < 0.05. MPO: Myeloperoxidase, BGD: Beta-Glucuronidase, NE: Neutrophil Elastase.

**Table 4 ijerph-18-08075-t004:** Correlations between Biochemical Parameters and Clinical Parameters in Smokers, Non-Smokers and all Participants.

Characteristics		*n*-Sm (*n* = 21)			Sm (*n* = 39)			Total (*n* = 60)		
		MPO	BGD	NE	MPO	BGD	NE	MPO	BGD	NE
PI	*r*	0.188	−0.338	−0.497	0.051	0.000	0.209	0.156	0.211	0.221
	*p*	0.415	0.134	* 0.022	0.757	0.999	0.203	0.235	0.106	0.090
GI	*r*	0.038	−0.404	0.049	−0.006	0.104	0.576	0.072	0.074	0.369
	*p*	0.869	0.069	0.834	0.973	0.528	* <0.001	0.582	0.572	* 0.004
BoP (%)	*r*	0.123	−0.359	−0.019	0.030	0.122	0.384	0.114	0.127	0.309
	*p*	0.596	0.110	0.936	0.855	0.459	* 0.016	0.387	0.334	* 0.016
PD (mm)	*r*	−0.140	0.125	−0.033	0.050	−0.160	0.308	0.031	0.007	0.250
	*p*	0.545	0.588	0.888	0.762	0.331	0.056	0.816	0.961	0.054
CAL (mm)	*r*	−0.133	0.110	−0.032	0.062	−0.102	0.346	0.039	0.024	0.275
	*p*	0.566	0.634	0.891	0.707	0.538	* 0.031	0.766	0.854	* 0.034

Calculated using Pearson’s Correlation Analysis, *r* = correlation coefficient. * Statistically significant at level of *p* < 0.05. MPO: Myeloperoxidase, BGD: Beta-Glucuronidase, NE: Neutrophil Elastase.

**Table 5 ijerph-18-08075-t005:** Logistic Regression Analysis of Biochemical and Clinical Parameters.

Characteristics	Beta	SE	*p* *	OR	%95 CI		% Equation
Smoking	2.996	0.730	0.000	20.000	4.780	83.687	16.827

* *p* < 0.05.

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
