# Peer review of "Impact of Smoking on Neutrophil Enzyme Levels in Gingivitis: A Case-Control Study"

_ijerph, 2021, doi:10.3390/ijerph18158075_

Round 1
Reviewer 1 Report
Dear authors, the article is very interesting, although it is not completely a novelty. Therefore, some concerns were raised (below).
- The study had 60 patients analyzed good number; ethical committee approved (ok).
- There is an incoherence in the study. The title and the inclusion criteria talked about gingivitis... and in the study was observed CAL, which is observed in periodontitis (exclusion criterion). I suggest to adequate it, although the result showed no difference (what is a logical result).
Line 103: Why was considered 5 cigarettes? And 12 months? In the current classification was considered 10.
Author Response
CORRESPONDENCE: There is an incoherence in the study. The title and the inclusion criteria talked about gingivitis... and in the study was observed CAL, which is observed in periodontitis (exclusion criterion). I suggest to adequate it, although the result showed no difference (what is a logical result).
RESPONSE: We included CAL measurements in our clinical examination to confirm diagnosis of subjects as gingivitis.
CORRESPONDENCE: Why was considered 5 cigarettes? And 12 months? In the current classification was considered 10.
RESPONSE: In the current classification it is considered 10 cigarettes per day for grading periodontitis. In our study we aimed to investigate more subtle changes in gingivitis. We selected at least 12 months of smoking consumption* to avoid including erratic smokers and to better characterize tobacco influence. Still this is a limitation of our study and we would like to thank you for the notice.
*Javed F, Abduljabbar T, Vohra F, Malmstrom H, Rahman I, Romanos GE. Comparison of Periodontal Parameters and Self-Perceived Oral Symptoms Among Cigarette Smokers, Individuals Vaping Electronic Cigarettes, and Never-Smokers. J Periodontol. 2017 Oct;88(10):1059-1065. doi: 10.1902/jop.2017.170197. Epub 2017 Jun 23. PMID: 28644108.
We thank the reviewer for his/her valuable comments.

Reviewer 2 Report
There are some needs to improve in the table presentation and statistical analysis of the results.
1) Tables presentation
I think it would be better to merge some of the tables.
Table 1+2 = Table 1
Table 3+4 = Table 2
Table 5 = Table 3
Table 6 = no need to present
Table 7 = Table 4
2) Statistical analysis of the results
(1) Some of the variables might show non-parametric distribution because we used the non-parametric analysis for PI, PD, CAL, MPO, and BGD. However, there was no mention of the results of the distribution of normality by the Kolmogorov-Smirnov test. Please mention the results of the distribution of the normality test (might be positioned the line 150).
(2) Please make clear about the independent and dependent variables (in the line between 154 to 156). The smoking status can not be a dependent variable. The biochemical and clinical parameters are changed because some of the patients were smokers which is the reason for the change. Generally dependent variable is the outcome and independent variables are selected as the related or effective factors for outcome variables.
(3) Please give more information in Table 7 (in line 243).
It looks BGD and PI which you presented in the Characteristics were outcomes, dependent variables as shown in line 156. Generally, the researchers put the independent variables in the Characteristics column. Please change the column name as outcomes or dependent variable rather than Characteristics. Also, the title can be changed such as 'Logistic regression analysis of biochemical and clinical parameters by the independent variable, smoking status'.
Please describe as well the reason why the other biochemical and clinical parameters such as GI or MPO did not show in Table 7. Also, write the reason to make the biochemical and clinical parameters as binary outcomes in the methods part.
It needs to write footnotes with adjustments with other variables such as demographic data as well.
Author Response
CORRESPONDENCE: I think it would be better to merge some of the tables.
Table 1+2 = Table 1
Table 3+4 = Table 2
Table 5 = Table 3
Table 6 = no need to present
Table 7 = Table 4
RESPONSE: Thank you for your contribution. We merged the Table 1 and Table 2 as you suggested. We removed Table 6 and described these results in the Results section.
Accordingly in the manuscript:
“Table 1. Demographic Data for Smokers and Non-Smokers with Gingivitis.
Characteristic |
Sm (n=39) |
N-Sm (n=21) |
Total (n=60) |
p value |
Age (years), mean ± SD |
21.28 ± 2.45 |
21.80 ± 2.29 |
21.46 ± 2.39 |
0.421* |
Gender, n (%) |
|
|
|
0.016† |
Males |
29 (74.4) |
9 (42.9) |
38 (63.3) |
|
Females |
10 (25.6) |
12 (57.1) |
22 (36.7) |
|
Education, n (%) |
|
|
|
0.704† |
< Bachelor degree |
5 (12.8) |
2 (9.5) |
7 (11.6) |
|
‡ Bachelor degree |
34 (87.2) |
19 (90.5) |
53 (88.4) |
|
Toothbrushing frequency, n (%) |
|
|
|
0.275† |
Twice a day or more |
24 (61.5) |
17 (81.0) |
18 (30.0) |
|
Once a day |
14 (35.9) |
4 (19.0) |
41 (68.3) |
|
Less than once a day |
1 (2.6) |
0 (0.0) |
1 (1.7) |
|
Smoking duration (year), n (%), mean ± SD |
4.76 ± 3.50 |
|
|
|
1-5 years, median, (IQR) |
28 (71.8) |
|
|
|
6-10 years, median, (IQR) |
8 (20.5) |
|
|
|
11-15 years, median, (IQR) |
3 (7.7) |
|
|
|
Cigarettes consumption (cig/day), n (%), mean ± SD |
14.74 ± 0.77 |
|
|
|
5-10 cig. per day, median, (IQR) |
8 (20.5) |
|
|
|
10-20 cig. per day, median, (IQR) |
14 (35.9) |
|
|
|
20+ cig. per day, median (IQR) |
17 (43.6) |
|
|
|
* Differences between groups were tested using Independent Samples T-test.
† Association between smoking and each independent variable were analyzed using the x2 test and Tukey’s post hoc analysis.”
Page 5
“GCF levels of MPO showed no significant correlation with GCF levels of BGD and NE (p>0.05), whereas GCF levels of BGD correlated positively with GCF levels of NE (r= 0.321, p=0.006)” Line 220
CORRESPONDENCE: Some of the variables might show non-parametric distribution because we used the non-parametric analysis for PI, PD, CAL, MPO, and BGD. However, there was no mention of the results of the distribution of normality by the Kolmogorov-Smirnov test. Please mention the results of the distribution of the normality test (might be positioned the line 150).
RESPONSE: Thank you for your contribution. Distribution normality was analyzed by Kolmogorov-Smirnov test. The results were mentioned as you requested.
Accordingly in the manuscript:
“GI, BoP, and GCF levels of NE showed normal distribution and PI, PD, CAL, and GCF levels of MPO, and BGD didn’t show normal distribution.” Line 150
CORRESPONDENCE: Please make clear about the independent and dependent variables (in the line between 154 to 156). The smoking status cannot be a dependent variable. The biochemical and clinical parameters are changed because some of the patients were smokers which is the reason for the change. Generally, dependent variable is the outcome and independent variables are selected as the related or effective factors for outcome variables.
RESPONSE: Thank you for your attention. Smoking status is defined as an independent variable. It is corrected in the manuscript.
Accordingly in the manuscript:
“Models were constructed by defining gender, smoking status, BoP, PI, PD, CAL, and GI scores as independent variables and GCF levels of BGD as the dependent variable. In order to estimate the probability of independent variables in terms of enzyme levels, binary logistic regression analysis (Forward LR) was performed.” Line 154-156
CORRESPONDENCE: Please give more information in Table 7 (in line 243).
It looks BGD and PI which you presented in the Characteristics were outcomes, dependent variables as shown in line 156. Generally, the researchers put the independent variables in the Characteristics column. Please change the column name as outcomes or dependent variable rather than Characteristics. Also, the title can be changed such as 'Logistic regression analysis of biochemical and clinical parameters by the independent variable, smoking status'.
Please describe as well the reason why the other biochemical and clinical parameters such as GI or MPO did not show in Table 7. Also, write the reason to make the biochemical and clinical parameters as binary outcomes in the methods part. It needs to write footnotes with adjustments with other variables such as demographic data as well.
RESPONSE: Three models were constructed by defining gender, smoking status, BoP, PI, PD, CAL, and GI scores as independent variables and GCF levels of MPO, BD, and NE as dependent variables. Models were constructed separately for each enzyme. Regarding the results of binary logistic regression analysis, the model which was defined by GCF levels of BGD as the dependent variable was considered the illustrative model of the study and the results were given in the manuscript. Additionally, the title of the Table 5 was reworded as suggested.
Accordingly in the manuscript:
“Smoking status explained the model. The probability of having GCF levels of BGD above 25.34 μg/ml in Sm has an OR=20.00 time (95%CI: 4.780-83.687) (Table 5).
Table 5. Logistic Regression Analysis of Biochemical and Clinical Parameters.
Characteristics |
Beta |
SE |
p* |
OR |
%95 CI |
% Equation |
|
Smoking |
2.996 |
0.730 |
0.000 |
20.000 |
4.780 |
83.687 |
16.827 |
*p<0.05” Page 7
We thank the reviewer for his/her helpful comments.
Reviewer 3 Report
The study by Omer-Cihangir et al., is a case study of gingivitis in smokers and non-smokers. The researchers recorded periodontal clinical parameters and the presence of neutrophil enzymes. A relationship between the presence of increased neutrophil enzyme concentration and smoking was seen. The authors are careful not to overstate their results. One of the limitations of the paper is that the data can’t answer if smoking causes an increase in neutrophils which in turn increases the concentration of neutrophil enzymes or that smoking may cause the neutrophils present to degranulate at a higher rate than neutrophils from nonsmokers. Being able to answer this question is beyond the scope of this paper. Other factors may be involved including the effect of smoking on the bacteria present and the increase in biofilm that could also cause an increase in inflammation, and neutrophil degranulation.
Minor revision.
Discussion
Line 248
Herein it was observed….
This sentence needs to be reworded. Higher enzymes don’t show increased inflammation. Higher inflammation was seen concurrent with higher enzyme levels, or clinical parameters of inflammation correlated with enzyme concentration. The data can only infer that one causes the other.
Author Response
CORRESPONDENCE: The study by Omer-Cihangir et al., is a case study of gingivitis in smokers and non-smokers. The researchers recorded periodontal clinical parameters and the presence of neutrophil enzymes. A relationship between the presence of increased neutrophil enzyme concentration and smoking was seen. The authors are careful not to overstate their results. One of the limitations of the paper is that the data can’t answer if smoking causes an increase in neutrophils which in turn increases the concentration of neutrophil enzymes or that smoking may cause the neutrophils present to degranulate at a higher rate than neutrophils from nonsmokers. Being able to answer this question is beyond the scope of this paper. Other factors may be involved including the effect of smoking on the bacteria present and the increase in biofilm that could also cause an increase in inflammation, and neutrophil degranulation.
RESPONSE: Thank you for your contribution. As you pointed out, since we do not have a group of healthy participants, we tried to only emphasize a possible association between smoking and gingivitis. We have added this into the discussion part. Due to the absence of a healthy group, the source of inflammation was not investigated in this manuscript.
Accordingly in the manuscript: “Since this study does not include a group of healthy participants, it has been only emphasized a possible association between smoking and gingivitis.” Line 298
CORRESPONDENCE: Discussion, Line 248, Herein it was observed…This sentence needs to be reworded. Higher enzymes don’t show increased inflammation. Higher inflammation was seen concurrent with higher enzyme levels, or clinical parameters of inflammation correlated with enzyme concentration. The data can only infer that one causes the other.
RESPONSE: Inflammatory response includes several mechanisms, including enzymatic activity. It is shown that the severity of inflammation results in higher enzymatic activity. Even though, higher enzyme levels don’t show increased inflammation. Our data showed that both enzyme levels and clinical parameters of gingival inflammation which are BoP and GI are higher in smokers. Our data imply that smokers have a tendency to show increased inflammatory activity. This sentence is rephrased in the manuscript. Thank you for your contribution.
Accordingly in the manuscript: “Herein, the data infer that the higher levels of neutrophil enzymes in GCF in Sm seem to associate with increased inflammation influenced by smoking.” Line 248
We thank the reviewer for his/her helpful comments
Round 2
Reviewer 2 Report
It is corrected enough to publish now.